# Relative Growth of Lettuce (*Lactuca sativa*) and Common Carp (*Cyprinus carpio*) in Aquaponics with Different Types of Fish Food

**George Vernon Byrd** [1,2,*] and **Bibhuti Ranjan Jha** [1]

1   Department of Environmental Science and Engineering, Dhulikhel Campus, Kathmandu University, Dhulikhel 45200, Nepal

2   College of Science and Technology, University of the Nations Kona Hawaii Campus, Box 75-5851, Kuakini Hwy, Kailua Kona, HI 96740, USA

*   Correspondence: byrdv2@gmail.com

**Abstract:** Aquaponics has the potential to contribute to food security in urban Nepal, where agricultural land near cities is rapidly being converted for other uses. This technology's use is expanding in Nepal, but the relatively high cost of commercial fish food is a hindrance. As a result, some aquaponics operators are resorting to alternative, less expensive fish foods. Since the primary input of nutrients to the plants grown in aquaponics comes from the fish food, an evaluation of the impact of fish foods on plant and fish growth is needed to help operators evaluate the costs and benefits of commercial compared to alternative fish diets. This study evaluated the growth of lettuce and common carp, the most common species of plant and fish used in aquaponics in Nepal, with three fish diets (commercial fish food, commercial chicken food, and a homemade diet with mustard oil cake and rice bran) at a commercial aquaponics farm with nine identical systems allowing for three replicates of the three fish food treatments. There were no significant differences in the measurements of lettuce growth (stem length, root length, and stem mass) and few differences in nutrient concentrations in leaf tissue. The specific growth rate of the carp fingerlings was lowest for the fish in the systems fed with the homemade diet (0.21) compared to those fed commercial fish food or commercial chicken food (0.31 and 0.28, respectively). These findings suggest that aquaponics operators who have been buying the more expensive commercial fish food with fish meal as its protein source can save 50–95% of the related costs by using commercial chicken food or the homemade diet defined in this study. This could potentially encourage the expansion of aquaponics systems in Nepal.

**Keywords:** aquaponics; common carp; fish food; growth; lettuce

## 1. Introduction

Aquaponics is a soilless growing technique that combines hydroponics and aquaculture. Instead of using inorganic chemicals in a nutrient solution to grow plants as in hydroponics, aquaponics uses nutrients obtained through the bioconversion of metabolic "wastes" from fish (or other aquatic organisms) and uneaten fish food [1–4]. These "closed" recirculating systems may have particular applications in urban agriculture because food can be grown in relatively high densities with high water and nutrient efficiency and with very little discharge to the environment [2]. The technology lends itself to controlled environments, and since no soil is needed, it can be deployed in or on buildings in cities [5].

Although the modern form of aquaponics is relatively new, it is being adopted widely, particularly in developed countries [6,7], and it has the potential to help with local food production in less-developed countries where rapid urbanization is reducing the available agricultural land near cities. For example, Nepal is experiencing one of the highest rates of urbanization in south Asia [8]. The space for growing fresh food near cities such as Kathmandu is disappearing rapidly [9,10]. Soilless growing in Nepal started in 2010, and at least 50 early adopters are currently trying aquaponics, often on flat roof tops and mostly

in the Kathmandu Valley [11]. Interviews with system owners indicated that the high cost of commercial fish food was considered a limitation, and many aquaponics operators have adopted less expensive alternatives [11].

The composition of fish food is important since it is the primary nutrient input to aquaponics systems [12,13]. As in Nepal, the cost of fish food is typically a major part of operating expenses for aquaponics everywhere [14]. Fish meal and fish oil have been the major sources of protein in fish food historically, but there is concern about the sustainability of this source, which has resulted in research into alternative inputs [15–18]. Commercial aquaculture is the primary market for fish food. Studies of fish diets have focused on fish production, fish health, and inputs that increase the nutritive quality of fish as food for humans [12,19–22]. In aquaponics, fish food must be adequate for fish, and it must also provide nutrients for the plants [12]. An understanding of the balance between fish-stocking rates, the amount of fish food used, and the number or plants in a system is important to optimize nutrient dynamics. As a result, the research on feeding rate ratios [2,23] includes numerous case studies addressing growth and other response variables for different combinations of plants, fish, and types of hydroponic grow beds [13,23–30]. Most of these studies have analyzed commercial fish food. Despite the discussions in the literature regarding alternate fish diets with less dependence on fish meal and fish oil as a protein source [12], there seems to be little, if any, research on the effects of non-commercial fish food on plant production in aquaponics [31].

To evaluate the question regarding the impact of alternate fish diets on plant and fish production in Nepal, we used the plant–fish combination of lettuce (*Lactuva sativa*) and common carp (*Cyprinus carpio*), the two most frequent species used in aquaponics in Nepal [11]. The objective of this study was to determine whether the high cost associated with commercial fish food was justified based on food production. We compared plant and fish growth and nutrients in the plant tissue in a replicated study using commercial corn-based chicken food and mustard oil cake/rice bran along with commercial fish food as treatments. The objective was to use an operational farm for the study so that the results could be applied in a Nepali context.

## 2. Materials and Methods

A commercial aquaponics farm near Pokhara, Nepal, was used for this experiment, which spanned from 10 November to 22 December 2021 for the plant growth portion and extended to 28 April 2022 for the fish growth section. The aquaponics units were situated in a passive greenhouse that was completely enclosed with a heavy plastic top, which extended over the screened sides and could be raised or lowered to help control air flow and temperature. Fans also helped with air flow. The floor space was 20 m × 6 m.

For the experiment, nine identical aquaponics units were employed. Each unit included a 200 L plastic barrel fish tank and two half barrels (100 L each) for the plant-growing beds. The surface area of each grow bed was about 0.46 m$^2$, so each pair of grow beds supported by the single fish tank totaled 0.92 m$^2$. Each grow bed was filled with irregularly shaped gravel ranging from 0.5 to 1.5 cm at maximum length. Bell siphons [32] were installed in each bed to flush water back into fish tanks after the beds were filled. Submersible SOBO brand 25 W water pumps with a capacity of 1800 L/h (made by Zhongshan Hongling Electrical Appliance Co., Ltd., Guangdong, China) were used to circulate water through 20 mm PVC pipes. Dissolved oxygen was generated by the vigorous flushing of water through the siphon to the water surface in the fish tank. However, each tank also had one air diffusion stone, with each pair of tanks sharing one 5 W, Sheng Zhe brand air pump (made by Shanghai Shengzhe Machinery Technology Company in Shanghai, China) diffusing 3.5 L/min. The supplemental oxygen from the air stones was meant to keep the fish alive if a siphon malfunction went unnoticed for an extended period (e.g., overnight). The maximum vertical head height when water in the fish tanks was at its lowest level was typically 90 cm. Water levels were maintained such that at the minimum level (when both

grow beds were full at the same time just before the siphons started) there remained at least 70–80 L of water in the fish tank. Full tanks contained about 150 L of water.

The three types of fish food used were those typically available to aquaponics operators in Nepal: (1) Commercial Fish Food (CFF), namely, KoiKing pellet; (2) Shakti brand G-2 Commercial Chicken Food (CCF); and (3) Homemade (H) from equal amounts of rice bran (about 12–16% protein [33–35] and mustard oil cake (with an estimated 27–28% protein [33])). Table 1 lists the ingredients in each diet, the results of proximate analysis conducted at the National Animal Feed, and typical costs.

**Table 1.** Primary ingredients, proximate composition, and costs of three types of fish food used in experimental aquaponics units.

|  | Commercial Fish Food | Commercial Chicken Food | Homemade |
|---|---|---|---|
| Ingredients [1] | Fish meal<br>Spirulina<br>Wheat germ<br>Krill powder<br>Yeast<br>Vitamins and minerals | Maize<br>Soya rice<br>Calcium powder<br>Grit<br>Vitamins and minerals | Mustard oil cake<br>Rice bran |
| Labeled Protein % Composition [2] | 35 | 25 | no label |
| Crude Protein | 35.7 | 18.2 | 22.5 |
| Crude Fat | 6.0 | 2.9 | 2.1 |
| Crude Fiber | 2.2 | 2.1 | 9.5 |
| Crude Ash | 9.1 | 4.6 | 9.0 |
| Cost | 800–1200 npr/kg [3] | 60–80 npr/kg | 60 npr/kg |

Notes: [1] For commercial fish food and chicken food, the ingredients were taken from the labels on bags. [2] Proximate Composition Analysis: Kjeldahl method was used for protein, while the others were analyzed with near-infrared spectroscopy. [3] This could be as low as 160–300 npr/kg for those close enough to buy in bulk from the feed mill at the Directorate of Fisheries in the Kathmandu Valley.

On 7 November 25–30, "Green Wave" leaf lettuce seedlings were transplanted at 12–14 days old directly into the gravel media of the nine system grow beds, in which half of the seedlings were placed in each of the two half drums. No plant tubes, net pots, or support meshes were used. Lettuce seedlings were thinned to 20 in each grow bed on 15 November when they were 4–5 cm tall.

Beneficial bacteria were well-established in the experimental systems because they were part of the aquaponics farm that had been operational for more than 6 months prior to this experiment, which had been growing lettuce with tilapia in the tanks. Therefore, we were able to stock tanks with 25–30 fingerling common carp on 10 November. The fish ranged roughly from approximately 6–9 cm, although only a small sample was measured, and none were weighted to avoid handling stress.

Each of the three fish food types was randomly assigned to three systems, and the assigned diets were fed to the fish twice daily throughout the study. The fish were fed the commercially formulated diets of commercial fish food and commercial chicken food (Table 1) and we formulated the homemade diet by mixing mustard oil cake (which had been turned to paste by hydration) with rice bran each at 50% by weight. The mixture was pressed into small balls and dried in the sun briefly before being fed to the fish. Fish were fed each morning and afternoon. The total amount fed was about 12–15 g per tank per day (approximately the amount the carp ate in 10 min and about 2% of their biomass as measured on 22 December 2021). This amount that was fed remained constant throughout the study.

Lettuce seedlings were grown for 45 days after being transplanted. Several measures were used to document lettuce response to the three fish food treatments. Stem length (longest leaf) was measured for each plant in mid-growth (1 December) and at harvest (22

December). Stem (including leaf) mass and root length were measured for each plant at harvest.

Leaf samples were collected, dried (with a Silva Homeline dehydrator, Silva Schneider Company, Salzburg, Germany) at the site, and analyzed for macro nutrients (N, P, K, Ca, Mg, and S) at the Agricultural Technology Center in Lalitpur, Nepal. Analysis techniques for various elements were as follows: N—Kjeldahl digestion distillation; P—Vanadomolybdate; K—flame photometric; Ca and Mg—dry ashing and EDTA; and S—wet digestion and turbidimetric measurement.

Water temperatures were measured with an Aquaneat digital thermometer (CLL Pet Supplies LLC, Madison, Wisconsin, US) daily in the early afternoon during the plant growth stage of the experiment and weekly thereafter through the fish growth study. Average water temperatures were identical among systems in November (19.4 °C) and nearly identical in December (17.6 °C) during the plant growth study. After the plant growth section of the experiment, and the finding that there were no differences among systems, less frequent measures were made with monthly averages in January–April of 15.2 °C, 16.8 °C, 22.2 °C, and 27.4 °C.

Water chemistry (pH, ammonia, nitrite, and nitrate) was measured in the field with API Freshwater Master Test Kits weekly during the plant growth component of the study but less frequently later. Phosphate levels were measured with a Fluval $PO_4$ test kit. General hardness and carbonate hardness were measured periodically with API 5-in-1 test strips. Indicators of water quality were similar among systems with different treatments (Table 2). Weekly measures of ammonia ($NH_4$) and nitrite ($NO_2$) rarely exceeded safe levels (<1.0 mg/L, [4]) for the fish and bacteria, only reaching 1 mg/L once in one system. The plant food, nitrate, was frequently present at slightly higher concentrations in the systems with CFF but was almost always at least 40 ppm in all systems, which is adequate for plant growth. Phosphate levels were most frequently 2.5 or 5 mg/L, and the water was hard with high levels of GH and KH, thereby buffering against change in pH, which usually remained at 8 or 8.2.

**Table 2.** A comparison of field-measured water chemistry parameters in aquaponics systems with three different fish food treatments.

| Treatment [1] | Statistic | $NH_4$ | $NO_2$ | $NO_3$ | $PO_4$ | GH [2] | KH [3] | pH |
|---|---|---|---|---|---|---|---|---|
| CFF | Mode [4] | 0.25 [5] | 0.25 | 80 | 2.5 | 180 | 120 | 8.0 |
|  | Max | 0.5 | 0.25 | 120 | 5 | 180 | 180 | 8.4 |
|  | Min | 0 | 0 | 20 | 1 | 180 | 80 | 7.4 |
| CCF | Mode | 0.25 | 0 | 40 | 5 | 180 | 120 | 8.2 |
|  | Max | 0.5 | 0.25 | 120 | 5 | 180 | 180 | 8.4 |
|  | Min | 0.25 | 0 | 20 | 1 | 180 | 120 | 7.4 |
| H | Mode | 0.25 | 0 | 40 | 2.5 | 180 | 120 | 8.2 |
|  | Max | 1.0 | 0.5 | 120 | 5 | 180 | 180 | 8.4 |
|  | Min | 0 | 0 | 10 | 1 | 180 | 80 | 7.8 |

Notes: [1] CFF=commercial fish food, CCF = commercial chicken food, and H = homemade. [2] GH = general hardness. [3] KH = carbonate hardness. [4] Mode was used instead of mean because these are ordinal data. [5] All measures are in ppm except for pH.

To assess carp growth, a longer period was needed than for plants. Fish were fed the prescribed diets from the initial stocking on 10 November 2021 until 28 April 2022 (168 days). Due to previous high mortality in fish introduced to the farm, which was likely due to stress during transport, the fish were not subjected to additional stress of weighing and measuring until 22 December 2021, when carp biomass averaged 702 ± 12.8 g per tank (no difference among treatments at 0.05 level). Individual carp averaged 17.0 ± 1.3 g and 9.2 cm. The period used for fish growth was 128 days. The hydroponic grow beds continued to produce lettuce throughout this period but were on a staggered schedule to meet farm customer needs, so the plant growth could not be comparably measured.

One way analysis of variance (with Tukey LSD multiple comparisons where appropriate) was used to test the null hypotheses, namely, that there was no difference in the plant, fish, and water response variables with respect to the three fish food treatments.

## 3. Results

### 3.1. Plant Growth

The average length of the longest lettuce leaves among treatments ranged from about 10–11 cm at mid-growth and from about 17–21 cm at the harvest stage (Table 3). There were no significant differences among the treatments at either stage.

**Table 3.** Lettuce growth among systems with three different fish food treatments.

| Variable | CFF [1] | | CCF | | H | | Difference |
|---|---|---|---|---|---|---|---|
| | Mean | SE [2] | Mean | SE | Mean | SE | |
| Leaf len. (cm) at mid-growth | 10.5 | 0.29 | 10.8 | 0.37 | 9.8 | 0.12 | NS [3] |
| Leaf len. (cm) at harvest | 20.2 | 0.94 | 17.1 | 1.01 | 20.9 | 1.21 | NS |
| Daily leaf growth [4] (cm) | 0.52 | 0.06 | 0.38 | 0.13 | 0.46 | 0.04 | NS |
| Stem and leaf wt. (g) | 37.1 | 4.46 | 34.2 | 7.71 | 34,8 | 6.30 | NS |
| Root len. (cm) | 9.4 | 0.47 | 10.4 | 0.69 | 9.5 | 0.70 | NS |

Notes: [1] CFF = commercial fish food, CCF = commercial chicken food, and H = homemade. [2] Standard error. [3] Differences not significant at 0.05 level. [4] DGR = average length at harvest-average length at mid-growth/number of days [36,37].

The average daily growth of the lettuce stems (as indicated by the longest leaf) between mid-growth and harvest (a period of 21 days) was highly variable among the systems as indicated by the relatively large standard errors for the CCF and H treatments (Table 3). The means were not significantly different.

As for daily growth, there was also a relatively high variation in the fresh mass of the individual plants, but the means, which ranged from about 34–37 g, were not statistically different among the treatments (Table 3).

The average lengths of the fresh roots varied from about 9.5–10.5 cm among treatments (Table 3). The differences were not significant at the 0.05 level.

### 3.2. Plant Leaf Nutrients

Based on the nutrient samples of the lettuce leaves, the CFF and CCF diets resulted in very similar nitrogen levels (about 5%), but the H treatment resulted in significantly lower levels than the others (Table 4). For phosphorous, the CCF diet produced a higher concentration (0.20–0.28%) in the lettuce leaves than the other fish food treatments. The potassium levels were similar for all treatments (treatment means about 3.7–4.0%) (Table 4). Calcium levels were higher in leaves of lettuce grown with the H diet than the CCF diet, but other comparisons were not significantly different. There were no differences in the magnesium and sulfur levels in leaves among the diets; however, the mean sulfur level in the CCF diet was much lower and likely not significantly so because of the high variation among the individual systems' means, as indicated by the high standard errors (Table 4).

### 3.3. Fish Growth

The average carp fingerling weighed 13–23 g in the various systems on 22 December 2021 (T1) when the growth study began. The fish fed CFF and CCF gained significantly more weight than the fish fed H (Table 5). The log-transformed data (often used to determine the growth rates of smaller fish, such as the ones used in this study) indicated that the degree of growth was similar between the CFF and CCF treatments. The fish fed the H diet grew significantly slower than those fed CFF, but not quite significantly less ($p = 0.13$) than CCF (Table 5). The percentage gain in the length of the carp fingerlings averaged the highest for CFF and lowest for H diets, but the differences were not significant (Table 5). The condition indices based on length–weight ratios were similar for CFF and

CCF, and both were significantly higher than that for H. The survival rates were high for all treatments, with no significant differences (Table 5).

**Table 4.** Macronutrient concentrations in lettuce leaves.

| Nutrient | CFF [1] | | CCF | | H | | Difference at 0.05 Level of Significance [3] |
|---|---|---|---|---|---|---|---|
| | Mean | SE [2] | Mean | SE | Mean | SE | |
| Nitrogen [4] | 5.14 | 0.07 | 5.11 | 0.09 | 4.56 | 0.18 | H < CFF = CCF |
| Phosphorous [5] | 0.21 | 0.01 | 0.28 | 0.02 | 0.20 | 0.02 | CFF > CCF = H |
| Potassium [6] | 3.65 | 0.18 | 3.73 | 0.14 | 4.13 | 0.27 | NS [7] |
| Calcium [8] | 45.0 | 2.60 | 37.5 | 1.50 | 51.8 | 1.30 | H > CCF, H = CFF, CFF = CCF |
| Magnesium [8] | 239.3 | 49.3 | 281.3 | 59.6 | 231.0 | 40.7 | NS |
| Sulfur [8] | 83.3 | 19.8 | 41.4 | 4.6 | 88.5 | 45.0 | NS |

Notes: [1] CFF = commercial fish food, CCF = commercial chicken food, and H = homemade. [2] Standard Error. [3] Differences based on ANOVA and Tukey LSD values less than 0.05; underlines indicate pairwise differences. [4] Units: Total Kjeldahl N%. [5] Units: P %. [6] Units: $K_2O$%. [7] Not significant at the 0.05 level. [8] Units: mg/kg.

**Table 5.** Average growth in mass and length of individual common carp among treatments.

| Statistic | CFF [1] | | CCF | | H | | Difference [2] |
|---|---|---|---|---|---|---|---|
| Relative Weight Gain [3] | Mean | SE [4] | Mean | SE | Mean | SE | |
| T1 [5] | 19.5 | 3.25 | 16.6 | 1.85 | 15.9 | 0.80 | |
| T2 [6] | 30.3 | 5.89 | 24.7 | 3.71 | 19.6 | 1.46 | |
| % Change | 53.4 | 5.59 | 47.1 | 6.60 | 30.4 | 3.09 | H < CFF = CCF |
| Specific Weight Gain [7] | | | | | | | |
| T1 | 2.9 [8] | 0.18 | 2.8 | 0.12 | 2.7 | 0.05 | |
| T2 | 3.4 | 0.22 | 3.17 | 0.2 | 3.0 | 0.06 | |
| % Change | 0.33 | 0.03 | 0.29 | 0.03 | 0.21 | 0.02 | |
| Length Increase | | | | | | | H < CFF, H = CCF, CFF = CCF |
| T1 | 9.6 | 0.96 | 9.2 | 0.41 | 8.9 | 0.16 | |
| T2 | 11.1 | 0.82 | 10.5 | 0.51 | 9.8 | 0.22 | |
| % Change | 16.7 | 3.59 | 13.7 | 1.70 | 10.3 | 0.68 | NS |
| Condition Index [9] | 1.58 | 0.03 | 1.58 | 0.02 | 1.45 | 0.03 | CFF = CCF > H |
| % Survival | 0.94 | 0.03 | 0.96 | 0.02 | 0.98 | 0.01 | NS |

Notes: [1] CFF = commercial fish food, CCF = commercial chicken food, and H = homemade fish food. [2] Differences based on ANOVA values and Tukey LSD values less than $p$ = 0.05; underlines indicate pairwise differences. [3] Relative growth = ((T2 − T1)/T1) × 100. [4] Standard error. [5] Average weight (g) of individual carp at T1 (22 December). [6] Average weight (g) of individual carp at T2 (28 April). [7] Specific growth rate ((lnWtT2 − lnWt T1)/Days × 100). [8] Natural Log of weight (g) at the beginning (T1) and end (T2) of the growth period. [9] Condition index calculated as 100 × Weight/Length [3].

## 4. Discussion

The main aim of this study in Nepal was to determine the effects of three fish diets (commercial fish food with fish meal as a protein source and two commonly used alternative diets s) on plant and fish growth and nutrient levels in plant leaves.

As indicated above, the search for protein inputs in commercial aqua feeds, other than fish-based products, is a major area of aquacultural research around the world because of the declining availability and increasing cost of fish meal and fish oil ([16,22,38,39]). Plant- and insect-based substitutes are being tested [16,39], but some of these ingredients have properties that reduce fish growth due to their reduced palatability, anti-nutritional properties, and lower amounts of some amino acids [12,40,41]. Nevertheless, fish farmers in many areas are using farm-produced feeds ([35,42]), with some of the most common being mustard oil cake and rice bran. For example, in Nepal, India, and Bangladesh, various

"Indian" carp (e.g., *Carpio, Ctenopharyngodon, Labeo, and Hypophthalmichthys*) are frequently fed a diet with these ingredients [35,43,44]. Some aquaponics operators in Nepal use this diet and others use commercial corn-based chicken food as alternatives to commercial fish food [11].

The amount of crude protein recommended for good fish growth (frequently designed to maximize growth) and health varies depending on the type of fish; for instance, carnivorous fish typically need higher protein content (38–50%) than omnivores and herbivores (25–38%) [12,19,42]. The typically formulated feeds for omnivorous carp in tanks and cages include soy and corn to supplement fish meal as a protein source [42], and the use of alternatives to commercial fish food with fish meal and oil are widespread.

The analysis of the three diets used in this study (Table 1) indicated that CFF was similar to the recommendations [42] for protein, fiber, and ash content, but was slightly low (6% vs. 7.8%) in terms of fat. Protein and fat were lower in CCF and H diets, and the H diet was higher than recommended in fiber.

The two diets used in this study as alternates for commercial fish food with fish meal and fish oil are not unique. Common in aquacultural research are evaluations of fish diets for carp with major ingredients such as corn (such as our CCF) and often soy as major components [45–51], and like our H, with rice bran and mustard oil cake [33,52–54]. Several variables make direct comparisons among studies of carp growth difficult, such as the exact components of the diets, the initial size of the fish, whether the fish are in a pond or a recirculating tank, the feeding rates, and the water temperatures. Nevertheless, some general conclusions seem consistent between this study and other studies. Carp growth was often similar between the diets including fish meal as the primary protein source and those where fish meal was substantially or entirely replaced with corn and often soy as well, but growth was often slower with rice bran and mustard oil cake. In this study, the lower growth with the H diet compared to CCF might have been at least partially due to the higher digestibility of corn crude protein than rice bran by carp [55].

### 4.1. Plant Growth

The growth of the Green Wave leaf lettuce in the aquaponics units used in this study was generally similar among the treatments. The growth rates measured in this study were not abnormal, although rigorous comparisons were not possible due to differences among other studies [28,37,56–60] regarding the types of plants and fish used, the lengths of the studies, the stocking densities of the plants and fish, the fish food types and the amounts fed, and the different types of aquaponics systems.

### 4.2. Nutrients in Plant Tissue

In this study, the most expensive diet (CFF) did not result in significantly higher concentrations in the leaf tissue of any of the primary macronutrients except for nitrogen, which was higher in the systems with the CFF diet than those with H (but was similar to CCF). There are many studies of nutrients in lettuce leaves in soil-based systems (e.g., [61–63] and sufficiency ranges have been recommended [64]. The results from this study indicated that the nitrogen concentration in the leaf tissue was within the ranges found in other studies, but both phosphorus and potassium were at the lower end of the ranges that have been recorded or recommended for soil-based lettuce production (Table 6). Aquaponics systems are known to sometimes be deficient in these two macronutrients, which are particularly important for crops producing fruit such as tomatoes, and which are often added as supplements for these crops [12,65–67]. In addition, the experimental systems in this study had relatively high pH, a factor that has been reported to negatively affect phosphorus uptake in plants in aquaponics [68]. The levels of calcium and sulfates were relatively low in our study compared to those in soil (Table 7).

**Table 6.** Nitrogen, phosphorus, and potassium concentrations (%) in lettuce leaves in this study compared to other studies.

| Type of Study | N | P | K | References |
|---|---|---|---|---|
| Recommended sufficiency levels in soil | 4.5–6.5 | 0.30–0.80 | 6.0–10.0 | [64] |
| Soil-based | 4.4 | 0.70 | 8.1 | [61] |
| Soil-based | 3.1–6.0 | 0.35–0.75 | 2.5–7.7 | [62] |
| Soil-based | 2.5–5.0 | 0.35–0.85 | 3.0–9.0 | [63] |
| Aquaponics | 4.3–4.7 | 0.9–1.1 | 9.8–11.0 | [69] |
| Aquaponics | | 0.55 | 2.46 | [59] |
| Aquaponics | 4.5–6.2 | 0.6–0.9 | 2.4–3.7 | [70] |
| Aquaponics | 4.6–5.1 | 0.20–0.28 | 3.7–4.1 | This study |

**Table 7.** Calcium, magnesium, and sulfur concentrations (%) in lettuce leaves in this study compared to others.

| Type of Study | Ca | Mg | $SO_4$ | References |
|---|---|---|---|---|
| Recommended sufficiency levels in soil | 1–2 | 0.36–0.75 | 0.2–0.6 | [64] |
| Soil-based | 0.4–1.1 | 0.2–0.5 | 0.2–0.4 | [62] |
| Soil-based | 1.0 | 0.3 | 0.3 | [63] |
| Soil-based | 1.3 | 0.39 | 0.21 | [61] |
| Aquaponics | 1.4–2.0 | 1.0–1.8 | | [70] |
| Aquaponics | 0.1 | 0.2–0.4 | 0.1 | This study |

*4.3. Nutrients in Aquaponics Water*

In this study, the water quality was excellent with respect to the concentrations of the two forms of nitrogen that are toxic to fish, ammonia and nitrite, which were within safe levels [4] and were even lower than or similar to values recorded by others [71–74]. The levels of nitrate, a desirable plant food, were equal to or greater than the minimum values suggested for aquaponics [23] and were like the values found in other studies [59,75,76]. The field-measured approximate phosphate levels in the aquaponics nutrient solution in this study were lower than the recommended optimal levels [23], despite being like those found in some other studies (e.g., [75]).

*4.4. Fish Growth*

The specific growth rates recorded for carp were at the low end of the range recorded in other studies cited (Table 8), most of which occurred in warmer temperatures. The growth rates of carp are positively correlated with temperature [77], and the optimum temperature range for the growth of carp is 24–28 °C [78]. Lower temperatures, such as those during most of our study, can result in reduced feeding and growth rates [79]. Despite the relatively slow growth, the condition indices based on the length to weight ratios were similar to those reported for common carp in lakes [80]. In addition, survival was relatively high in our researched systems [81,82].

**Table 8.** Specific growth rates (SGR) of carp in this study compared to other studies.

| Type of Study | Fish Food [1] | Initial Wt (g) | Days of Study | Water Temp. (C) | SGR [2] | Reference/Comments |
|---|---|---|---|---|---|---|
| Tanks | FM, S, C | 40–49 | 42 | | 0.15 | [83] |
| Tanks | C, S | 13.5 | 56 | 16 | 0.57 | [45] |
| Pond | C | 333 | 210 | | 0.12 | [51] |
| Tanks | C | 11.5 | 63 | 25 | 0.70 | [48] |
| Tanks | FM, C, S | 115 | 126 | 27 | 1.02 | [46] |

**Table 8.** *Cont.*

| Type of Study | Fish Food [1] | Initial Wt (g) | Days of Study | Water Temp. (C) | SGR [2] | Reference/Comments |
|---|---|---|---|---|---|---|
| Tanks | FM, C | 10 | 56 | 25 | 1.79 | [47] chromium added |
| Tanks | R, M | <1 | 90 | 29 | 1.02 | [53] pea pod added |
| Pond | R, M | <1 | 330 | 27 | 0.51 | [84] polyculture |
| Tanks [3] | FM | 141 | 58 | 24 | 0.60 | [74] |
| Tanks [3] | FM | 50–70 | 52 | | 0.24 | [85] pH varied |
| Tanks [3] | FM | 36 | 70 | 27 | 0.39 | [25] fish species varied |
| Tanks [3] | FM | 40 | 30 | 23 | 0.77 | [86] symbiotic added |
| Tanks [3] | FM | 13–23 | 128 | 15–22 | 0.33 | This study |
| | C, S | 13–19 | | | 0.29 | This study |
| | R, M | 13–16 | | | 0.21 | This study |

Notes: [1] Primary components: FM = fish meal, S = soy, C = corn, R = rice bran, and M = mustard oil cake. [2] Specific growth rate = ((ln final wt-ln initial wt)/total days)) $\times$ 100. [3] Aquaponics; all others are aquaculture systems.

The relatively low growth rate of the carp in the tanks fed with the H diet might have resulted from lower acceptance. Although similar amounts of food were provided for all three diets, less aggressive feeding was observed in the fish given the H diets, and more uneaten food accumulated in these tanks than those where the other diets were provided.

## 5. Conclusions

This study demonstrates that lettuce production was not significantly lower in inexpensive alternative fish diets compared to commercial fish food. However, the carp grew more slowly on the homemade diet of rice bran and mustard oil cake than on commercial fish food with fish meal, but the growth rates when the fish were fed commercial corn-based chicken food were like those of commercial fish food. Nevertheless, the growth rates on all diets were relatively slow during the study, which was likely due to low water temperatures.

Most aquaponics operators do not have access to the bulk production of commercial fish food from government feed mills; therefore, choosing more widely available corn-based chicken food would reduce costs substantially (up to 95% for fish food from aquarium stores and more than 50% for fish food at government feed mills). The homemade diet is slightly cheaper, but this includes the labor employed to combine the ingredients. Even if reduced fish growth was not an issue (for operators concentrating mainly on plant production), the accumulation of waste rice bran in the system requires more maintenance. Since most aquaponics operators in Nepal grow lettuce and common carp, this finding should be broadly applicable and could potentially encourage others to adopt aquaponics where the cost of fish food is an impediment.

**Author Contributions:** Both authors were involved in the conceptualization and development of methodology. G.V.B. acquired the resources, collected the data, conducted data analysis, and prepared the original draft. B.R.J. provided supervision, insight through and review and editing of the manuscript. All authors have read and agreed to the published version of the manuscript.

**Funding:** This research received no external funding.

**Data Availability Statement:** Data are available from the senior author.

**Acknowledgments:** Matthew Farrell, owner of Farrell's Farm and Aquaponics, kindly allowed the research to be conducted at his farm. We acknowledge helpers at the aquaponics farm: Ajay Thami, Ajay Pariyar, Bharat Pariyar, and Manoj Chhetri. Drs. Smriti Gurung and Nani Raut at Kathmandu University offered helpful suggestions on data analysis and manuscript preparation.

**Conflicts of Interest:** The authors declare no conflict of interest.

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
