# Peer review of "Relative Growth of Lettuce (Lactuca sativa) and Common Carp (Cyprinus carpio) in Aquaponics with Different Types of Fish Food"

_water, doi:10.3390/w14233870_

Round 1

Reviewer 1 Report

Although the topic of proper feed selection for aquaponics is important, the study is not well done. I suggest rejection of the paper as the methodology doesn't comply with the basics of the fish feeding trials. One of the main rules for such a study to be valid is at least doubling the weight of the cultured fish during the experiment. In the current study, the yield of fish weight is just in arrange of 1-30 %, which isn't sufficient. Most likely, the fish were cultured on a vitamin and mineral deficient diet without a balanced amino acid profile of protein. recommendations from such a study would lead to an unethical culture of fish which would suffer nutritional deficiencies.

Author Response

I believe the low water temperatures were an important limiter in growth rates.  Thank you for your comments.  I went back and calculated condition indices for each treatment, and added that information in Table 5.  I believe this is only a general index that can be difficult to interpret, but these values suggest that at least the commercial fish food and commercial chicken food (which had significantly higher indices than the homemade diet) produced fish that were generally healthy (if I understand the index correctly) even though they grew slowly through the winter period of this study.  

Reviewer 2 Report

Authors should explain how the plants were grown (unclear), whether they had any tubular structures or root support mesh. 20 plants in 0.46 m² is a relatively high density, even for lettuce (what is the spacing between plants). Was there a comparison with lettuce grown in soil, using this water as reuse (in the same density and cultural management)? There is no need for the Number of the experimental system on the tables as the numbering gets confusing. Furthermore, as the differences between means are made by comparisons between treatments, this would allow the authors to join Tables 2 to 6 (as well as in the other sub-items of the Results) avoiding textual repetition. This would also be valid for Tables 7 and 8 that address nutritional characterization. What is the justification for using “mode” and not “average” in Table 10?

Would there be the possibility of continuing lettuce cultivation after 45 days (with a new transplant)? In this condition, it would probably be necessary to reduce the number of carp per tank, but depending on the cost-benefit ratio, the system could be further optimized.

After attending to these explanations, the article can be accepted.

Author Response

Thank you for the excellent recommendations.  

Point 1. Clarify how plants were grown: Lines 132 and 133 include clarification.

Point 2. Each system had nearly 1 m sq growing space (2 grow beds per system) so the density was about 20 plants/m sq. This was clarified in line 132.

Point 3. We did not have a comparison with lettuce grown in the soil.

Point 4. Reducing main tables to just treatment means and joining tables.  This has been done for all the tables as suggested, reducing the number of tables from 16 to 7.

Point 5.  Mode was used because the field measured water chemistry was ordinal data (We added a footnote in the table to explain)  

Point 6.  Unfortunately, the commercial facility that allowed the research is not available for adding another round of lettuce growth measurement with the three treatments. See lines 166-167.

Reviewer 3 Report

1.     Please rewrite Lines 114~121.

2.     “3.3 Water Characteristics” could be list in “Materials and Methods”.

3.     Lines 247~248: The water temperature changed a lot during the growth trial. Greenhouse did not work well? Water temperature should be kept in a narrow range for scientific research work.

4.     In Table 13, Please list the initial fish number and the initial fish body weight in the table.

Please list the fish survival rate in each barrel.

5.     Why the fish growth so slow in the experiment list in Table 13?

6.     Please keep 4.1 into Materials and Methods.

Author Response

Thank you for your thoughtful comments.

Point 1:  The section was rewritten, see lines 126-132.

Point 2. Water temperatures were moved to the Materials and Methods section as suggested, lines 156-175.    

Point 3.  Agreed, ideally the water temperature would not vary as much as it did in this study, but our intent was to conduct the research on a working farm typical of the types of aquaponics operations in Nepal and these passive screen houses do not have heat or cooling other than fans.

Point 4.  Table 13 was removed as the measures of individual fish growth were seen as clearer comparisons.  However, fish survival rate was added as suggested (Table 1). 

Point 5.  It turns out there were some mistakes in Table 13 and individual average fish growth was a clearer representation of the results. 

Point 6.  Done see Table 1.

Reviewer 4 Report

Higher fish stocking densities and the use of higher water temperatures, would elevate the interest in this and simillar studies,

Author Response

You make an excellent point.  We were restriced to the best season for lettuce production in the area of the study.  While good for lettuce, the low water temperatures caused slow growth of carp.  

Reviewer 5 Report

This is a valuable addition to the literature on aquaponics systems in developing countries.  Practitioners around the world are challenged by the high cost of formulated diets and transportation to their farms.  There are several revisions that should be considered prior to publication.

1.  Equipment used in the course of this study are not fully described.  The common description includes the city, state and country of the manufacturer.

2.  Initially, in text, diets were not fully described in terms of macronutrient concentrations, but are later in the paper as a Table in the Results section.  Suggest describing the diets in the Methods section.

3.  The approach and timing of system cycling prior to experimentation should be described.

4.  The nutritional history of the fish should be provided.  Fed the fish formulated diet?

5.  The statement "....food not varied...." is unclear.  Not varied as a function of body weight of the fish?  Or, as an amount per day?

6.  Citations should be provided for chemical analyses.

7.  Was dissolve oxygen not measured?

8.  Average initial weigh of fish should be provided in the Methods section.

9.  The statistical presentation is different from other papers like this one.  Most papers will list the accepted level of significance (LOS, for example p<0.05).  This paper presents the LOS for each measure.  I suppose this is technically correct and perhaps even acceptable, but it is an atypical presentation that is a bit difficult to follow.

10.  The authors state that root mass and length are highly correlated, but their r2 value is 0.49.  I personally do not consider a correlation coefficient of 0.49 as a high correlation.  

11.  Micronutrient concentrations are presented as mg/L, which is equivalent to mg/kg, but these measures were clearly measured as a mass not a volume.  Suggest changing to mg/kg.

12.  There are alot of Tables in this manuscript.  Suggest deleting those Tables in which data were not significantly different.  Those data could be presented in text.

13.  Chemical analysis of diets was presented in Table 17.  How were these values determined?  

Author Response

Thank you very much for your very helpful review.

Point 1.  Equipment manufacturer.  Names and locations of manufacturers of pumps were added, lines 86 and 91. 

Point 2.  Diet information was moved to as suggested.  See Table 2 line 103.

Point 3.  Cycling Description. Clarification was added. See line 131-133

Point 4. Fish feeding was clarified. See lines 135-142.

Point 5. Feeding rate remaining constant.  Clarification was added. See line 145.

Point 6. Reference for chemical analysis.  Lab is identified in line 153.

Point 7. Dissolved oxygen was not measured, but the flow rates and timing of siphon activation was measured in all systems to confirm that the water surface perturbation from siphon flushing was similar among systems and each system had one air diffusion stone, each pair of systems sharing on air pump.  Therefore, even though we did not have a DO meter, it is very likely that all the systems had very similar DO.

Point 8.  The average estimated length, without handling fish is listed in line 136.  As stated we only measured a few fish at the extremes and we did not weigh them to avoid the stress of handling immediately after transport.  Initial measuring and weighing of individual fish was on Dec. 22.

Point 9.  Listing statistical significance.  We reduced the multiple references when tables were consolidated.

Point 10.  The r square for our sample was significant at the 0.05 level, but you are correct of course that 0.49 is not an extremely strong correlation coefficient.  We removed that statement but continued to omit the root mass because after thinking about this, it was clear that fresh mass had some other problems in an ebb and flow system, such as how recently the root had been saturated in water.

Point 11.  Covert mg/l to mg/kg.  I checked with the lab that did the analysis and they infact meant mg/kg and that has been changed.

Point 12.  Another reviewer suggested combining a lot of what was in original tables and this accomplished the reduction you suggested.

Point 13. This table was moved to Materials and Methods at the suggestion of another reviewer and the lab is mentioned line 102 as well as the methods they used in lines  122-123

Round 2

Reviewer 1 Report

Unfortunately, the corrections canť change the fact that the experiment isn't well done. I suggest rejection of the paper as the methodology doesn't comply with the basics of the fish feeding trials. One of the main rules for such a study to be valid is at least doubling the weight of the cultured fish during the experiment. In the current study, the yield of fish weight is just in arrange of 1-30 %, which isn't sufficient. Most likely, the fish were cultured on a vitamin and mineral-deficient diet without a balanced amino acid profile of protein. recommendations from such a study would lead to an unethical culture of fish which would suffer nutritional deficiencies. Such a study shouldn't be published with such a design. Diets have to fulfill the nutritional requirements of the cultured fish, environmental conditions should be within optimal range, and fish has to at least double their weight.

Author Response

As indicated in my earlier response, the objective was to use the diets that are being used in aquaponics in Nepal currently to try to understand the consequence on plant and fish growth of diets other than commercial fish food.  Growth was slow during the study on all diets which probably represents what happens in all systems using these same diets during winter.  In defense of the approach, the feeding rate was about 2% of biomass, which is what is recommended, there was almost no mortality, and the condition indexes were normal.  I added a statement in conclusions to acknowledge slow grow rates on all diets (lines 371-372), including the commercial diet which is specifically balanced for Cyprinidae, so it seems likely that slow growth was related primarily to winter water temperatures and not to nutrition in formulated diets.    

Reviewer 3 Report

Please list P value in Table 5.

Author Response

A p value of <0.05 was considered a significant differences and that is now reflected in footnote 2 of Table 5.